# Exposure to Triclosan and Bisphenol Analogues B, F, P, S and Z in Repeated Duplicate-Diet Solid Food Samples of Adults

**DOI:** 10.3390/toxics9030047

**Published:** 2021-03-03

**Authors:** Marsha K. Morgan, Matthew S. Clifton

**Affiliations:** 1United States Environmental Protection Agency’s Center for Public Health and Environmental Assessment, Research Triangle Park, NC 27711, USA; 2United States Environmental Protection Agency’s Center for Environmental Measurement and Modeling, Research Triangle Park, NC 27711, USA

**Keywords:** adults, phenolic compounds, consumer products, food packaging, diet, exposure, intake dose, risk assessment

## Abstract

Triclosan (TCS) and bisphenol analogues are used in a variety of consumer goods. Few data exist on the temporal exposures of adults to these phenolic compounds in their everyday diets. The objectives were to determine the levels of TCS and five bisphenol analogues (BPB, BPF, BPP, BPS, and BPZ) in duplicate-diet solid food (DDSF) samples of adults and to estimate maximum dietary exposures and intake doses per phenol. Fifty adults collected 776 DDSF samples over a six-week monitoring period in North Carolina in 2009–2011. The levels of the target phenols were concurrently quantified in the DDSF samples using gas chromatography/mass spectrometry. TCS (59%), BPS (32%), and BPZ (28%) were most often detected in the samples. BPB, BPF, and BPP were all detected in <16% of the samples. In addition, 82% of the total samples contained at least one target phenol. The highest measured concentration of 394 ng/g occurred for TCS in the food samples. The adults’ maximum 24-h dietary intake doses per phenol ranged from 17.5 ng/kg/day (BPB) to 1600 ng/kg/day (TCS). An oral reference dose (300,000 ng/kg/day) is currently available for only TCS, and the adult’s maximum dietary intake dose was well below a level of concern.

## 1. Introduction

Bisphenol A (BPA) is a high-production volume chemical that has been used in a variety of consumer goods, including food packaging (i.e., lining of metal cans) [1,2]. The use of BPA in food packaging has decreased in the last several years as concerns have been raised about human exposures and potential adverse health effects and has led to recent restrictions in the United States (US), Canada, and Europe [1,3,4]. Due to these restrictions, manufacturers have been increasingly replacing BPA in food packaging with other structurally similar lipophilic analogues such as bisphenol B (BPB), bisphenol F (BPF), bisphenol P (BPP), bisphenol S (BPS), and bisphenol Z (BPZ) [1,4,5,6]. Research has suggested that the diet is a major source of human exposure to these BPA replacement analogues [1,7,8]. However, limited data are available on the concentrations of BPB, BPF, BPP, BPS, and/or BPZ residues in commercially available foodstuffs, worldwide [2,5,7,8,9,10,11,12]. In the only published US study, Liao and Kannan [5] reported finding BPB, BPF, BPP, BPS, and/or BPZ residues in ~45% of the 267 canned and noncanned food items purchased from grocery stores in New York between 2008–2012. Among these five bisphenols, BPS (21%) and BPF (10%) were the most frequently detected in that study. This above information suggests that people are likely being exposed to several different bisphenol analogues in their daily diets in the US. Emerging research has indicated that exposure to these BPA replacement analogues may be adversely impacting human health (i.e., endocrine, metabolic, reproductive, and neurotoxic effects) [1,13].

Triclosan (TCS) is a chlorinated, phenolic compound that is lipophilic and is used as an antimicrobial agent in a wide variety of consumer goods, including household, personal care, and in clothing [14,15,16]. In the US, there are no known uses of TCS in food packaging, but it is incorporated as a material preservative in certain types of kitchenware (e.g., plastic utensils, cutting boards, and storage containers) and kitchen countertops that are intended to come in contact with foods [14,17]. There is growing concern about the potential dietary exposures of people to TCS residues resulting from the indirect use of some of these consumer goods (i.e., kitchenware, countertops, and personal care products) containing this phenol [14,16]. Currently, only three published studies conducted in China and Spain have measured TCS residue concentrations in any foods or beverages, and none in the US [18,19,20]. These small studies have reported finding TCS residues in one boiled ham sample (~2 ng/g) and in three raw chicken egg samples (up to 6.7 ng/g), but not in any cereal-based foods (i.e., corn flakes, pasta, cookies, and white bread) purchased from local supermarkets. In addition, Canosa et al. (2008) showed that slices of ham and cheese placed on a commercially available kitchen cutting board, impregnated with TCS, absorbed substantial residue levels (>40 ng/g) within five minutes of surface contact. This above information suggests that people’s everyday use of some common consumer goods containing TCS may be indirectly contaminating their foods with this chemical prior to consumption. Recent research has indicated that exposures to TCS may be causing adverse health effects (i.e., reproductive, thyroid, and on the gut microbiome) in humans [21,22,23].

In previous dietary exposure studies, researchers have collected food items from grocery stores or supermarkets and then analyzed these food items for specific phenols in a laboratory setting [2,5,7,8,9,10,11,12,18,19,20]. A major limitation of this approach is that it does not account for the possible additional contamination of these food items with phenolic compounds by people in real world settings. These types of real-life studies are necessary as previous research has suggested that the personal behavior of adults (i.e., use of certain types of kitchenware or personal care products on hands when preparing or eating foods) can considerably increase the levels of phenols in some foods prior to consumption [1,18,19,24].

Our “Pilot Study to Estimate Human Exposure to Pyrethroids Using an Exposure Reconstruction Approach” (Ex-R study) is the only study in the literature that has assessed the dietary exposure of adults to any phenol (BPA) in actual foods prepared and/or eaten in their real-life environments (i.e., home, work, or restaurants). In the Ex-R study, BPA residue levels were found in 38% of the 776 duplicate-diet solid food (DDSF) samples (maximum = 138 ng/g) collected by 50 adults over a six-week monitoring period in North Carolina (NC) 2009–2011 [24]. We are presently unaware of any published study that has reported the concentrations of other bisphenol analogues or TCS in foods consumed by adults in these everyday settings. In this current work, the objectives were to determine BPB, BPF, BPP, BPS, BPZ, and TCS residue concentrations in the 776 DDSF samples of Ex-R adults and to estimate their maximum dietary exposures and dietary intake doses to each phenol. Table 1 presents the chemical structures of the six phenolic compounds measured in the DDSF samples.

## 2. Materials and Methods

### 2.1. Study Background

The Ex-R study was originally designed to assess the temporal and cumulative exposures of adults to pyrethroid insecticides over a six-week monitoring period in their everyday settings [25]. Briefly, this exposure measurements study was performed at the US Environmental Protection Agency’s (EPA) Human Studies Facility in Chapel Hill, NC and within 40-miles of this facility at the participant’s homes in 2009–2011. In this study cohort, there were a total of 20 males, ages 19–48 years old, and 30 females, ages 21–50 years old. As a component of this study, the 50 adult participants filled out 24-h food diaries and collected 24-h DDSF samples on days 1 and 2 during sampling weeks 1, 2, and 6. Up to three separate DDSF samples were collected per participant each sampling day. A total of 776 DDSF samples were collected by the participants over the six-week monitoring period. In 2015–2016, the archived DDSF samples (*n* = 776) were removed from US EPA laboratory freezers (−20 °C) and quantified for the concentrations of BPB, BPF, BPP, BPS, BPZ, and TCS.

The University of North Carolina’s Institutional Review Board approved the Ex-R study design and sampling procedures (study number 09-0741) in August of 2008. All adult volunteers read and signed an informed consent form prior to participating. In addition, the 50 adults signed a second inform consent form permitting the archived DDSF samples to be analyzed for additional chemicals commonly found in residential settings.

### 2.2. Collection of Food Diaries and DDFS Samples

The collection of the food diaries was discussed in-depth in Morgan et al. [25]. The adult participants collected 24-h food diaries on days 1 and 2 of sampling weeks 1, 2, and 6. The participants filled out a paper copy of the 24-h diary over three consecutive sampling periods (period 1 [4:00–11:00 a.m.], period 2 [11:00 a.m.–5:00 p.m.], and period 3 [5:00 p.m.–4:00 a.m.]) each sampling day. These time periods were selected to represent when people generally ate breakfast, lunch, and dinner in the US. For each sampling period in the diary, the participants wrote with a provided ball point pen each food item (e.g., hamburger, pizza, apple, cake, soup, and smoothie) that they ate and checked a box that this item was also part of the corresponding DDSF sample (described below).

The collection of the DDSF samples was described earlier in Morgan et al. [25]. DDFS samples were defined as duplicate amounts of all food items, except beverages, that were consumed by a person during each sampling period. As part of their normal daily diets, the adult participants purchased their own solid food items during this study. The adult participants collected 24-h DDSF samples over three consecutive sampling periods each sampling day (as mentioned above). For each sampling period, participants placed identical amounts of each solid food item they ate into a resealable, polyethylene sampling bag (31 cm × 31 cm Uline Shipping Supply Specialist^®^, Pleasant Prairie, WI, USA). Then, this food sampling bag was placed into another resealable polyethylene bag (31 cm × 31 cm) for double containment. The food sampling bags were temporarily stored by the participants in provided portable thermoelectric coolers (34 cm L × 30 cm W × 36 cm H, Vinotemp^®^ or Princess International^®^, Los Angeles, CA, USA).

The participant returned the coolers, containing the food sampling bags and food diaries, to the Human Studies Facilities between 8:00 a.m. and 11:00 a.m. on Day 3 of each sampling week. At this facility, a technician checked in all study items with each participant and recorded the food mass (g) of each sampling bag (range = 10.6–1568 g) using a calibrated weight scale. The technician transported the coolers containing these study items with blue ice by motor vehicle to a US EPA laboratory about 20 miles away in Research Triangle Park, NC. At the laboratory, the DDSF samples (*n* = 776) were homogenized individually using a vertical cutter mixer (R10-Ultra^®^ or Robot Coupe R4N-D^®^) or a high-speed blender (WaringMBB518^®^) for high-liquid content foods like soups or fruit smoothies [25]. The samples were then aliquoted into 30 mL amber glass jars (12.0 g amounts each). The glass jars containing the homogenized food samples were stored in −20 °C laboratory freezers until chemical analysis.

### 2.3. Chemical Analysis of the DDSF Samples

At the laboratory, the 12.0 g aliquots for the DDSF samples (*n* = 776) were removed from the freezers and thawed overnight in refrigerators [25]. The six target phenols were concurrently extracted from each homogenized food sample using a QuEChERS (quick, easy, cheap, effective, rugged, and safe) method that was modified for complex food matrices [24,26,27]. Briefly, a 2 g aliquot of each homogenized food sample was weighed into a 50 mL polypropylene centrifuge tube, and 50 µL of an internal standard solution containing TCS ^13^C_12_, BPA ^13^C_12_, and BPS ^13^C_12_ (30 ng/g) was added. A ceramic homogenizer and 12 mL of acetonitrile was added to the tube and vortexed (2 min). Next, a magnesium sulfate and sodium acetate extraction salt (1 g) was added to the tube and vortexed (1 min) and centrifuged at 4000 RPM (5 min). The solvent layer was poured into a 15 mL polypropylene centrifuge tube containing 0.4 g of graphitized carbon black sorbent, and vortexed (1 min) and centrifuged (5 min). After that, the solvent layer was poured into a 50 mL conical glass tube and evaporated just to dryness using a parallel evaporator. The extract was reconstituted by adding 1.0 mL of acetonitrile to the tube, vortexed (5 s), and transferred to an autosampler vial. Finally, 100 µL of a solution of 99:1 trimethylsilyl, 2,2,2-trifluoro-*N*-(trimethylsilyl)-acetimidate was added and then vortexed (~20 s), heated at 80 °C (15 min), and cooled (15 min) to silylate the extract. Since BPA and analogues are a common contaminate in the laboratory, supplies were evaluated for their presence prior to use. Glassware was baked in a muffle furnace at 450 °C for 8-h to remove potential contaminates. In addition, reagent blanks were prepared with each sample batch in order to evaluate background levels that may be present in solvents and consumables as well as any potential contamination from equipment used during sample preparation.

The chemical analysis of each sample extract (1 mL) for BPB, BPF, BPP, BPS, BPZ, and TCS was performed using a gas chromatography/mass spectrometer equipped with an autosampler (Agilent 6890/5973, Agilent Technologies, Palo Alto, CA, USA) (see Table A1 of Appendix B) [24]. MassHunter Quantitative Analysis software was used for data processing (Agilent Technologies, Palo Alto, CA, USA). Separation was performed using a VF-Xms column (20 m × 0.15 mm × 0.15 µm film thickness, Agilent Technologies, Palo Alto, CA, USA) with a helium carrier gas at a constant flow rate of 0.4 mL/min. The capillary injector was used in splitless mode with split flow at 60 mL/min at 0.75 min. The injector temperature was 250 °C with a 2 µL injection volume into a single-gooseneck Ultra Inert glass liner (Agilent Technologies, Palo Alto, CA, USA). The thermal gradient started at 75 °C for 2 min to 217 °C at 10 °C/min, then to 223 °C at 1.0 °C/min, then to 330 °C at 20 °C/min, and holding for 5 min for a total run time of 37.5 min. The MS transfer line was maintained at 300 °C, with a source temperature of 230 °C and quadrupole temperature of 150 °C. The mass selective detector (MSD) was operated in selected ion monitoring (SIM) mode with electron impact (EI) ionization. The ions monitored represent the silylated molecules where the hydroxyl groups are replaced by trimethylsilyl (TMS) groups. Excellent sensitivity and selectivity were achieved since the molecular ion plus TMS for each chemical were able to be used for detection and quantification. Detailed parameters for ions collected with retention times are described in Table A2 of Appendix B. The method qualifiable limit (MQL) was 0.3 ng/g for each analyte in food, except for BPZ (0.5 ng/g). The target analytes were quantified with Agilent MassHunter software using least squares regression with a series of standard solutions. The standard solutions were in pure acetonitrile and were analyzed between sample extracts to compensate for and minimize matrix effects. Acceptance criteria for calibration was >0.99 correlation coefficent (r). Calibration was performed at a range of 0.5 to 1000 ng/mL (0.25 to 500 ng/g), but most sample data were found to be less than 100 ng/mL, so the calibration for concentration calculation was truncated to 0.5 to 100 ng/mL (0.25 to 50 ng/g). For compounds in samples that exceeded this range, the regression line was extended using the acquired calibration data as necessary.

### 2.4. Quality Assurance and Quality Control

In this study, the QC samples consisted of reagent blanks, matrix spikes (25 ng/g), and recovery spikes (25 ng/g). Among the six target phenols, there was only slight contamination of TCS (mean/standard deviation = 0.3 ± 0.6 ng/g) in the reagent blanks (*n* = 47), therefore, no background correction made. Extraction efficiency was calculated by dividing the found concentration by the expected concentration in the spiked samples. The result was multiplied by 100 to express the value in percent recovery. The mean percent recoveries for the 49 matrix spikes ranged from 92.3–133% for all analytes. For the 49 recovery spikes, the mean percent recoveries were between 93.9% and 113% for all analytes.

### 2.5. Statistical Analysis of the Data

For each target phenol, all sample data values below the MQL were assigned the value of MQL divided by the square root of 2 [28]. Summary statistics including detection frequencies, percentiles (50th, 75th, 95th, and 99th), and maximum values were calculated for the individual phenols in all the DDSF samples using GraphPad Prism, version 5.04 (GraphPad Software, San Diego, CA, USA). In addition, the participants’ estimated maximum dietary exposure (ng/day) and dietary intake dose (ng/kg/day) to each phenol was computed using the method described previously in Morgan et al. [29]. Only female participants had the highest maximum concentration of each phenol in the DDSF samples over a 24-h sampling period. The ages of the female participants were between 29 and 50 years old and their body weights ranged from 57.2–99.2 kg. Food mass records (g) were missing for two of the participants. For these two adults, the total food mass for each DDSF sample was estimated by using the recorded amount eaten (i.e., 3 cups) in the corresponding food diaries.

## 3. Results

### 3.1. Phenolic Compound Levels in the DDSF Samples

The concentration data values (ng/g) for the six phenolic compounds in each food sample are provided in Appendix A. Table 2 presents the summary statistics for the concentrations of BPB, BPF, BPP, BPS, BPZ, and TCS in the 776 DDSF samples of 50 Ex-R adults over a six-week monitoring period. The phenols detected the most often in the samples were TCS (59%), BPS (32%), and BPZ (28%). For TCS, median concentrations were 0.8 ng/g, and the maximum value was 394 ng/g. The maximum values for BPS and BPZ were 103 and 136 ng/g, respectively. Among these three phenols, the results showed at the 75th percentile that TCS levels (2.3 ng/g) were at least two times higher than BPS levels (0.6 ng/g) or BPZ levels (1.1 ng/g) in the samples. The other three phenols (BPB, BPF, and BPP) were detected in less than 16% of the samples. The maximum values were 2.3 ng/g (BPB), 217 ng/g (BPF), and 35.0 ng/g (BPP).

For the three most frequently detected phenols, Figure 1 presents the percentage of the participant’s DDSF samples with detectable concentrations of TCS, BPS, or BPS (>MQL) by sampling time period 1 (breakfast), period 2 (lunch), and period 3 (dinner). The results showed that the three different phenols were all detected in the DDSF samples the most often in period 3 and the least often in period 1.

### 3.2. Co-Occurrence of the Phenolic Compounds in the DDSF Samples

Figure 2 shows the co-occurrence of the six phenolic compounds in the 776 DDSF samples. The results showed that 82% of the total samples had one or more of these phenols. Specifically, in Figure 2, 40% of all the samples contained one phenol, and 29% of all the samples contained two different phenols. Additionally, 11% and 3% of the samples contained three and four different phenols, respectively.

At the subject level, the results showed that 10% of the participants had a minimum of three different phenols occurring in five or more of their DDFS samples over the six-week monitoring period. For these five participants, BPS, BPZ, and TCS were found together the most often in these samples, and residues ranged from <0.3–14.0 ng/g (BPS), <0.5–57.9 ng/g (BPZ), and <0.3–10.9 ng/g (TCS). In addition, two of these participants had at least eight DDFS samples that contained a minimum of three different phenols, and BPS, BPZ, and TCS co-occurred the most frequently in these samples. For these two male participants, the phenolic residue levels in the samples ranged from <0.3–14.0 ng/g (BPS), <0.5–57.9 ng/g (BPZ), and <0.3–6.7 ng/g (TCS).

### 3.3. Estimated Maximum Dietary Exposure and Dietary Intake Dose to Each Phenol

Table 3 presents the adult’s maximum concentration (ng/g), dietary exposure (ng/day) and dietary intake dose (ng/kg/day) to each phenol over a 24-h sampling period. The results showed that the participant’s maximum concentrations of the individual phenols ranged from 1.1 (BPB) to 220 ng/g (TCS) in the DDSF samples. The participant’s highest maximum 24-h dietary exposure was for TCS (120,690 ng/day) followed by BPZ (51,410 ng/day). Similarly, the greatest maximum 24-h dietary intake dose for these participants was for TCS (1600 ng/kg/day) and BPZ (706 ng/kg/day).

## 4. Discussion

In the last several years, BPA replacement analogues (i.e., BPB, BPF, BPP, BPS, and BPZ) have been increasingly used in food packaging and in other types of consumer goods (i.e., personal care products), worldwide [1,4,5,6,16]. Limited data are currently available on the concentrations of these alternative bisphenol analogues in foodstuffs purchased from grocery stores or supermarkets, and none in foodstuffs prepared and/or consumed by people in their everyday settings (e.g., home) [2,5,7,8,9,10,11,12,30]. Real world studies are especially needed as research has indicated that people’s personal behavior (i.e., microwaving meat in plastic containers, using the liquid fillings of canned vegetables, or handling thermal receipt papers before eating fast-food) can substantially increase bisphenol analogue residues in foods prior to consumption [31,32,33]. These previous studies performed in other countries (Canada, China, Portugal, and Spain) have reported finding residues of one or more of these alternative bisphenol analogues in canned and non-canned food items [2,7,8,9,10,11,12]. In a recent study conducted by Gonzalez et al. [2] BPB residues were found in canned chicken and olive oil (maximum = 3.9 ng/g) and in noncanned chicken and olive oil (maximum = 4.2 ng/g) bought from a large grocery store in Spain. Additionally, Cao et al. [8] detected BPF and BPS residues in > 45% of the 151 canned foodstuffs (i.e., meats, vegetables and fruits) purchased from supermarkets in China between 2017–2018, and maximum residues were 75.4 ng/g (BPF) and 1.6 ng/g (BPS). In the only US based study, Liao and Kannan [5] detected BPB, BPF, BPP, BPS, and/or BPZ residues in ~45% of the 267 food items bought from local grocery stores in New York 2008–2012. In that study, the five bisphenols were found mainly (>90%) in the food samples, and BPS (21%) and BPF (10%) were detected the most often. The highest maximum residue for BPS was in an unspecified meat product (23.8 ng/g) and for BPF in a mustard dressing sample (1130 ng/g). The authors reported finding no significant differences in the levels of these bisphenol residues in canned foods compared to noncanned foods (i.e., glass, paper, and plastic). In comparison to Liao and Kannan [5], our current study also quantified the concentrations of these five bisphenol analogues in 776 DDSF samples of 50 adults in NC between 2009–2011. Our results showed that one or more of these bisphenols was detected in 82% of the total DDFS samples. BPS (32%) and BPZ (28%) were detected the most frequently in the samples. The maximum value for BPS of 103 ng/g was found in a participant’s sample containing a cheese and tomato sandwich. For BPZ, the maximum value of 136 ng/g occurred in a participant’s sample containing an apple, spinach salad, tuna, quinoa, and cereal bar. However, BPF had the highest measured residue level of 217 ng/g (in a bologna and cheese sandwich) found in all the DDSF samples in the Ex-R study. Our data results agree with Liao and Kannan [5] indicating that the measured bisphenol residues likely originated from a variety of different foods stored in both canned and noncanned food packaging in the US between 2008–2012. As the composition of these alternative bisphenol analogues in food packaging likely changes overtime, additional research is needed to determine the exposures of adults to these chemicals in their everyday diets.

TCS is not known to be incorporated into any food packaging materials [17], but this chemical has been used in a variety of consumer goods such as personal care products and kitchenware [14,15,16,17]. Only a few studies in the literature have quantified the concentrations of TCS residues in any foodstuffs, globally [18,19,20]. In a recent study by Azzouz et al. [20], TCS residues were not detected in 13 different cereal-based foods (cookies, corn flakes, macaroni, muesli with fruit, multiseed bread, noodles, rice, sesame reganus, spaghetti, tortellini with cheese, wheat flour, wheat tortillas, and white bread) that were purchased from several shops and supermarkets in Spain. However, in another study by Yao et al. [19], TCS residues were found in three raw chicken eggs (maximum = 6.7 ng/g), but not in samples of beer, soda, chicken meat, or cherries purchased at supermarkets in China. The authors suggested that the eggshells may have been indirectly contaminated at the supermarkets by employees using products (e.g., sponges or wipes) containing TCS to clean food preparation surfaces. In comparison to these other studies, our results showed that TCS residues were detected in 59% of the 776 DDSF samples consumed by 50 Ex-R adults in NC. The median TCS concentration was 0.8 ng/g and 10.0 ng/g at the 95th percentile. In addition, a female participant had the two highest measured concentrations of TCS (394 and 152 ng/g) found in two different DDSF samples eaten on the same sampling day at her home. One sample contained a fruit bar and a banana (394 ng/g of TCS), and the second sample contained a chicken breast, baked beans, coleslaw, and a biscuit (152 ng/g of TCS). All the other collected DDSF samples (*n* = 9) by this participant contained TCS residue levels of less than 6.5 ng/g. We suspect that this participant’s unknown personal behavior (e.g., use of kitchenware or a skin care product containing TCS) likely attributed to the much higher TCS residue levels (>151 ng/g) in these two DDSF samples. This is plausible as research has shown that measurable levels of TCS residues (>40 ng/g) can migrate from kitchenware (cutting board) into foods within five minutes of contact [18]. In addition, TCS was found in >72% of the 114 personal care products (i.e., body washes, skin lotions, toilet soaps, hair care, and makeup) commonly used by US adults in 2012–2013 [16]. Furthermore, research has indicated that the consumption of tap water in prepared and/or cooked foods is likely not an appreciable source of adult exposure to TCS in the US [34]. It is important to mention that the Ex-R study was performed prior to the 2016 US Food and Drug Administration’s (FDA’s) ban of TCS in all antiseptic wash products that are used with water including hand soaps, bar soaps, and body washes [35]. It is currently unknown if the ingestion exposures (dietary and non-dietary) of adults to TCS have substantially decreased after the 2016 ban by the US FDA, and more research is warranted. In addition, there is currently a void of data in the literature on the levels of other antimicrobial agents commonly used in consumer goods like kitchenware that can potentially migrate info foods prior to consumption.

A few published studies conducted before 2002 have quantified the co-occurrence of several different phenolic compounds in DDSF samples of children, but none of adults [36,37,38]. Our study results showed that 29% of the 776 DDFS samples of Ex-R adult participants contained two different target phenols, and TCS and BPS were found together the most often in these samples. In addition, 11% of the total samples had at least three different target phenols, and TCS, BPS, and BPZ co-occurred the most frequently in these samples. More research is needed on the phenolic compounds that co-occur the most often in foodstuffs commonly eaten by the general US population. This information would be beneficial to help elucidate the possible major sources (e.g., certain types of consumer goods) of adult exposures to phenols in consumed foods.

In this current work, we estimated the Ex-R adult’s maximum dietary intake doses to BPB, BPF, BPP, BPS, BPZ and TCS via consumed solid foods over a 24-h sampling period (Table 3). Only female participants had the highest maximum dietary intake doses per phenol, and 60% of these females (ages 29–36 years old) were non-Hispanic black. Based on the available data, we could not find a definitive reason (i.e., eating certain foods) why these specific female participants had the highest maximum concentrations of each phenol occurring in the 24-h DDSF samples in this study. For these five participants, we suspect that their unknown use of certain types of consumer products (e.g., personal care products on hands or kitchenware) while preparing and/or eating foods likely contributed to the elevated phenolic concentrations occurring in these food samples Assuming 100% absorption in the gut, the participant’s maximum dietary intake doses were the highest for TCS (1600 ng/kg/day) followed by BPZ (706 ng/kg/day). For these six target phenols, the US EPA has an established oral reference dose (RfD) for only TCS of 300,000 ng/kg/day [17]. This information showed that the adult’s maximum dietary intake dose to TCS was 187 times lower than the established oral RfD by the US EPA. Currently, oral RfDs are not available for BPB, BPF, BPP, BPS and BPZ in the US or other countries [39]. Therefore, we could not ascertain whether the maximum dietary intake doses of the Ex-R participants to the individual bisphenol analogues were below a level of concern.

Few data are currently available on the aggregate exposures of adults (dietary, nondietary, inhalation, and dermal) to these newer BPA replacement analogues, and none for TCS [40]. In a recent 2020 study, Karrer et al. [40] suggested that both dietary and nondietary sources of BPS or BPF were likely important in the aggregate exposures of 144 Norwegian adults, but other relevant sources (i.e., actual food sample concentration data) were likely missing in this assessment. As data are limited, more research is needed to elucidate the important sources and routes of human exposure to these BPA replacement analogues and to TCS.

In conclusion, the Ex-R adults were exposed to levels of BPB, BPF, BPP, BPS, BPZ and TCS residues in a variety of solid foods they consumed over a six-week monitoring period. Among the target phenols, only TCS was detected frequently (>55%) in the 776 DDSF samples. As there are no known uses of TCS in food packaging in the US [17], these TCS data suggest that over half the food samples consumed by the Ex-R adults were indirectly contaminated by unknown sources (e.g., consumer products) of this chemical prior to consumption.

## Figures and Tables

**Figure 1 toxics-09-00047-f001:**
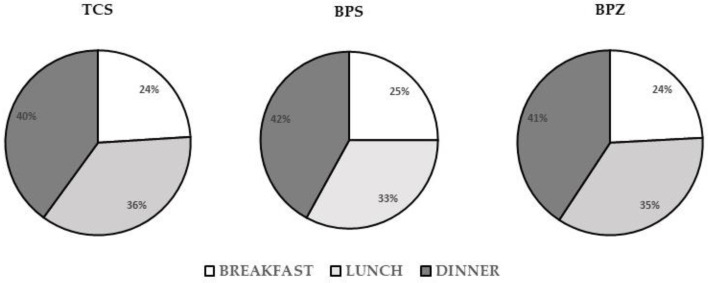
Percentage of DDSF samples with detectable levels of TCS, BPS, and BPZ residues by sampling time period 1 (breakfast = 4:00–11:00 a.m.), period 2 (lunch = 11:00 a.m.–5:00 p.m.), and period 3 (dinner = 5:00 p.m.–4:00 a.m.).

**Figure 2 toxics-09-00047-f002:**
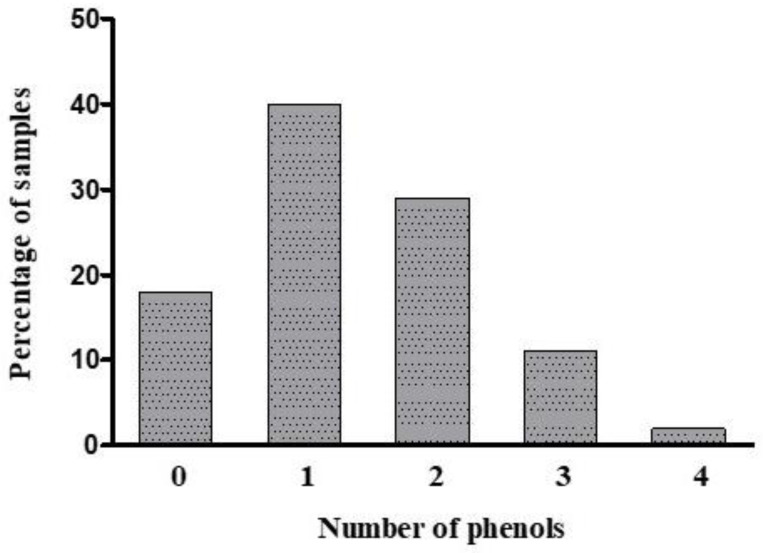
Co-occurrence of the target phenols in the 776 DDSF samples.

**Table 1 toxics-09-00047-t001:** Chemical structures of the six target phenols.

Phenol	Acronym	CAS Number	Structure
Bisphenol B	BPB	77-40-7	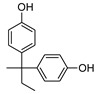
Bisphenol F	BPF	620-92-8	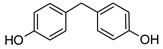
Bisphenol P	BPP	2167-51-3	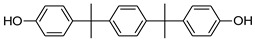
Bisphenol S	BPS	80-09-1	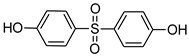
Bisphenol Z	BPZ	843-55-0	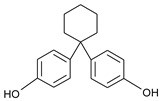
Triclosan	TCS	3380-34-5	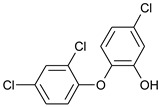

**Table 2 toxics-09-00047-t002:** Phenolic compound levels (ng/g) in 776 DDSF samples.

Phenol	% ^a^	Percentiles				Maximum
		50th	75th	95th	99th	
BPB	1	--- ^b^	---	---	---	2.3
BPF	15	---	---	9.5	57.2	217
BPP	4	---	---	---	1.5	35.0
BPS	32	---	0.6	3.4	21.2	103
BPZ	28	---	1.1	6.8	22.8	136
TCS	59	0.8	2.3	10.0	34.3	394

^a^ The frequency of detection of a phenol was > the method quantitation limit (MQL). ^b^ Data value was below the MQL for a phenol.

**Table 3 toxics-09-00047-t003:** The adults’ estimated maximum concentration, dietary exposure and dietary intake dose to each phenol over a 24-h sampling period.

Phenol	24-h Level ^a^ (ng/g)	Dietary Exposure (ng/Day)	Dietary Intake Dose (ng/kg/Day)
BPB	1.1	1220	17.5
BPF ^b^	46.2	29,390	296
BPP ^bc^	7.0	4500	59.8
BPS	33.3	13,640	238
BPZ	81.3	51,410	706
TCS ^bc^	220	120,690	1600

^a^ Highest concentration of a phenol measured in a participant’s 24-h DDSF sample. ^b^ Mass of food (g) was estimated using the amount recorded in the participant’s food diary. ^c^ One female participant had the highest levels of both BPP and TCS in their 24-h DDSF samples.

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
