# Peer review of "Exposure to Triclosan and Bisphenol Analogues B, F, P, S and Z in Repeated Duplicate-Diet Solid Food Samples of Adults"

_toxics, 2021, doi:10.3390/toxics9030047_

Round 1

Reviewer 1 Report

The topic of the article Exposure to Triclosan and Bisphenol Analogues B, F, P, S, and Z in Repeated Duplicate -Diet Solid Food Samples of Adults is very original and clear. This is an excellent report and I recommend its publication. The literature review and methodology were painstakingly thorough and incorporated the use of sufficient numbers of samples in the study. It would be better that authors give possible explanations of the elevated maximum concentrations of each phenol in the DDSF samples of female participants.

Author Response

Reviewer 1

Comments and Suggestions for Authors:

The topic of the article Exposure to Triclosan and Bisphenol Analogues B, F, P, S, and Z in Repeated Duplicate -Diet Solid Food Samples of Adults is very original and clear. This is an excellent report and I recommend its publication. The literature review and methodology were painstakingly thorough and incorporated the use of sufficient numbers of samples in the study. It would be better that authors give possible explanations of the elevated maximum concentrations of each phenol in the DDSF samples of female participants.

Response: We have added the following sentences to the fourth paragraph of the Discussion section: “Based on the available data, we could not find a definitive reason (i.e., eating certain foods) why these specific female participants had the highest maximum concentrations of each phenol occurring in the 24-h DDSF samples in this study. For these five participants, we suspect that their unknown use of certain types of consumer products (e.g., personal care products on hands or kitchenware) while preparing and/or eating foods likely contributed to the elevated phenolic concentrations occurring in these food samples.”

Reviewer 2 Report

This study is overall well presented and mehodologically sound.

Nevertheless, before the study can be recommended for publication, the Authors should give consideration to the following points:

  • the study has the value of a "screening", pointing out the overall dietary exposure in adults and evaluating whether a concern might exist or not, in comparison with the available toxicologically reference values. The study cannot identify any food commodity/category as a major determinant of intake. If such food commodity/commodities do exist, then high consumers of that/these commodities would represent a special high-exposure group
  • The study does not consider children, which is a limitation to be acknowledged
  • For compounds like Triclosan and also bisphenols, the dietary intale is a component of a broader aggrtegate exposure. For instance, EFSA (2015) evaluated that non-dietary sources were important in the aggregate exposure of adolescents and adults to Bisphenol A.. Are there data on the relative contribution of dietary and non-dietary sources for Triclosan and/or the investigated Bisphenols ?
  • While all compounds examoned are "phenols", the Discussion/Conclusions should be made clear that Triclosan and Bisphenols are well different in regard of a) routes of food contamination, b) toxicological mechanisms and modes of action
  • The authors should discuss whether Bisphenols may have common mode(s) of actions and targets, and whether a risk assessment should consider the cumulative exposure to bisphenols (including the widespread Bisphenol A) 
  • Finally, both Triclosan and Bisphenols are potential developmental toxicants due to their endocrine-disrupting actions. The Authors should discuss the potential for short-term high intakes based on their data, as these might be hazardous if they occur during susceptible pregnancy windows

Author Response

Reviewer 3

Comments and Suggestions for Authors:

This study is overall well presented and mehodologically sound.

Nevertheless, before the study can be recommended for publication, the Authors should give consideration to the following points:

  • the study has the value of a "screening", pointing out the overall dietary exposure in adults and evaluating whether a concern might exist or not, in comparison with the available toxicologically reference values. The study cannot identify any food commodity/category as a major determinant of intake. If such food commodity/commodities do exist, then high consumers of that/these commodities would represent a special high-exposure group

Response:  We agree that we could not identify any food commodity or food category as a major determinant of the participants’ intake to the individual phenols because our DDSF samples were composited over three, consecutive time periods each sampling day. But more importantly, our study has indicated that the personal behavior of adults (e.g., use of certain types of consumer products containing a phenol(s) while preparing/eating foods) likely substantially increased the levels of a phenol(s) in some foods prior to consumption. So, in real world settings, human behavior is probably a major driving factor that is substantially increasing the phenolic compound residue levels in some foods before consumption, particularly at the high end of exposure.  

We have added a new paragraph in the Introduction section to discuss this important point we made above: “In previous dietary exposure studies, researchers have collected food items from grocery stores or supermarkets and then analyzed these food items for specific phenols in a laboratory setting {2,5,7-12,18-20] . A major limitation of this approach is that it does not account for the possible additional contamination of these food items with phenolic compounds by people in real world settings. These types of real-life studies are necessary as previous research has suggested that the personal behavior of adults (i.e., use of certain types of kitchenware or personal care products on hands when preparing or eating foods) can considerably increase the levels of phenols in some foods prior to consumption [1,18,19,24].”   

  • The study does not consider children, which is a limitation to be acknowledged

Response: We disagree. In the Materials and Methods section, we clearly state that only adults participated in the Ex-R study.  It is unclear why we would need to acknowledge another group (i.e., children) that was not part of this study?

  • For compounds like Triclosan and also bisphenols, the dietary intale is a component of a broader aggrtegate exposure. For instance, EFSA (2015) evaluated that non-dietary sources were important in the aggregate exposure of adolescents and adults to Bisphenol A.. Are there data on the relative contribution of dietary and non-dietary sources for Triclosan and/or the investigated Bisphenols?

Response:  We have added a new paragraph to the discussion section: Few data are currently available on the aggregate exposures of adults (dietary, nondietary, inhalation, and dermal) to these newer BPA replacement analogues, and none for TCS [40]. In a recent 2020 study, Karrer et al. [40] suggested that both dietary and nondietary sources of BPS or BPF were likely important in the aggregate exposures of 144 Norwegian adults, but other relevant sources (i.e. actual food sample concentration data) were likely missing in this assessment. As data are limited, more research is needed to elucidate the important sources and routes of human exposure to these BPA replacement analogues and to TCS.

  • While all compounds examoned are "phenols", the Discussion/Conclusions should be made clear that Triclosan and Bisphenols are well different in regard of a) routes of food contamination, b) toxicological mechanisms and modes of action

Response:  We have modified the first sentence in the first paragraph of the discussion section: In the last several years, BPA replacement analogues (i.e., BPB, BPF, BPP, BPS, and BPZ) have been increasingly used in food packaging and in other types of consumer goods (i.e., personal care products), worldwide [1,4-6,16]. We also added a new sentence to the second paragraph of the discussion section: TCS is not known to be incorporated into any food packaging materials [17], but this chemical has been used in a variety of consumer goods such as personal care products and kitchenware [14-17].

We did not report any internal measurements or biomarkers such as urine in this manuscript, so a discussion of the toxicological mechanisms and modes of action of the phenols in humans is outside the current scope of this work.

  • The authors should discuss whether Bisphenols may have common mode(s) of actions and targets, and whether a risk assessment should consider the cumulative exposure to bisphenols (including the widespread Bisphenol A) 

Response:  We did not report any internal measurements or biomarkers such as urine (only external measurements – food) in this manuscript, so a discussion of the toxicokinetics/toxicodynamics including common modes of action and target tissues (organs) for these chemicals is outside the current scope of work.

Although outside this current scope of work, we do agree that future risk assessments should consider the cumulative exposures of humans to these bisphenol analogues (including BPA) from all sources and routes. But, we first need to develop such basic things like oral reference doses --- which are lacking in the literature for all of the target bisphenols, except for BPA (we already mention this important information on the lack of oral reference doses for the target bisphenols in the discussion section).

  • Finally, both Triclosan and Bisphenols are potential developmental toxicants due to their endocrine-disrupting actions. The Authors should discuss the potential for short-term high intakes based on their data, as these might be hazardous if they occur during susceptible pregnancy windows

Response: As part of the eligibility criteria and IRB constraints for the Ex-R study, pregnant women were not included in our study cohort. So, we do not have data on the intake doses of these phenols for pregnant women – which may be quite different (e.g., eating behaviors) than are current study cohort. Therefore, it would not be appropriate to make assumptions about the short-term, high intakes of pregnant women to these phenols in this current manuscript.

Reviewer 3 Report

The present work reports results of the assessment of exposure to Triclosan and Bisphenol Analogues B, F, P, S and Z in Repeated Duplicate-Diet Solid Food Samples of Adults. I recommend this manuscript for publication in Toxic after improving certain aspects of the manuscript.

  1. Elements of scientific novelty should be presented in a detailed and convincing manner (in the last paragraph of the Introduction). In addition, it should also be briefly described in the Abstract.
  2. The current importance of the field should be clearly given in the Introduction.
  3. I suggest that a diagram (scheme) presenting the step of the to Materials and Methods sub-section.
  4. Remember to report amounts of Rs with the corresponding errors (value ± SD). Otherwise, contents are meaningless. This should be done for data in figures and in tabels and text.
  5. How extraction efficiency was calculated? It should be presented in the text.

Author Response

Reviewer 4

Comments and Suggestions for Authors:

The present work reports results of the assessment of exposure to Triclosan and Bisphenol Analogues B, F, P, S and Z in Repeated Duplicate-Diet Solid Food Samples of Adults. I recommend this manuscript for publication in Toxic after improving certain aspects of the manuscript. 

1. Elements of scientific novelty should be presented in a detailed and convincing manner (in the last paragraph of the Introduction). In addition, it should also be briefly described in the Abstract.

Response: We agree and have added a new paragraph in the Introduction section: In previous dietary exposure studies, researchers have collected food items from grocery stores or supermarkets and then analyzed these food items for specific phenols in a laboratory setting {2,5,7-12,18-20]. A major limitation of this approach is that it does not account for the possible additional contamination of these food items with phenolic compounds by people in real world settings. These types of real-life studies are necessary as previous research has suggested that the personal behavior of adults (i.e., use of certain types of kitchenware or personal care products on hands when preparing or eating foods) can considerably increase the levels of phenols in some foods prior to consumption [1,18,19,24].

Due to the maximum word limit (200 words) for the current abstract, we could not add a condensed version of this information in it without deleting other more important information or data.

2. The current importance of the field should be clearly given in the Introduction.

Response:  We agree and have revised the last paragraph in the Introduction section as follows: Our “Pilot Study to Estimate Human Exposure to Pyrethroids Using an Exposure Reconstruction Approach” (Ex-R study) is the only study in the literature that has assessed the dietary exposure of adults to any phenol (BPA) in actual foods prepared and/or eaten in their real-life environments (i.e., home, work, or restaurants). In the Ex-R study, BPA residue levels were found in 38% of the 776 duplicate-diet solid food (DDSF) samples (maximum = 138 ng/g) collected by 50 adults over a six-week monitoring period in North Carolina (NC) 2009–2011 [24]. We are presently unaware of any published study that has reported the concentrations of other bisphenol analogues or TCS in foods consumed by adults in these everyday settings. In this current work, the objectives were to determine BPB, BPF, BPP, BPS, BPZ, and TCS residue concentrations in the 776 DDSF samples of Ex-R adults and to estimate their maximum dietary exposures and dietary intake doses to each phenol. Table 1 presents the chemical structures of the six phenolic compounds measured in the DDSF samples.

3. I suggest that a diagram (scheme) presenting the step of the to Materials and Methods sub-section.

 Response:  We disagree.  In the Materials and Methods section, we have already included a citation #25 (Morgan et al. Temporal variability of pyrethroid metabolite levels in bedtime, morning, and 24-hr urine samples for 50 adults in North Carolina. Environ. Res. 2016, 144, 81–91.) that provides an in-depth discussion of the Ex-R study design and sampling methodology. Also, in this citation, it contains a detailed sampling diagram that includes the collection of the food samples and food diaries. Furthermore, we have provided a comprehensive summary of the collection of the food samples and food diaries in the text in this section of the manuscript.

4. Remember to report amounts of Rs with the corresponding errors (value ± SD). Otherwise, contents are meaningless. This should be done for data in figures and in tabels and text.

Response: We disagree. The frequencies of detection for the target phenols in the DDSF samples were less than 33%, except for TCS (59%). Therefore, it is very appropriate that we have used the current summary statistics (detection frequency, 50th, 75th, 95th, 99th, and/or maximum) for the levels of each target phenol measured in the food samples in the provided tables and figures.  In addition, we have presented the same/similar summary statistics on the levels of chemicals found in DDSF food samples (in both tables and graphs) in several previously published dietary exposure articles (for instance, citations #24 “Morgan et al. Distribution, variability, and predictors of urinary bisphenol A levels in 50 North Carolina Adults over a six-week monitoring period. Environment International. 2018, 112, 85–99 and #29 Morgan et al., Pyrethroid insecticides and their environmental degradates in repeated duplicate-diet solid food samples of 50 adults.  Journal of Exposure Science and Environmental Epidemiology. 2018: 28: 40-45. – none of the reviewers have had any issues in the way these food data were presented in these published articles.

5. How extraction efficiency was calculated? It should be presented in the text.       

Response: We have added the following two sentences to subsection 2.4 of the Material and Methods section: Extraction efficiency was calculated by dividing the found concentration by the expected concentration in the spiked samples. The result was multiplied by 100 to express the value in percent recovery.

Reviewer 4 Report

Review:

Manuscript Number: Toxics 355698

Title: Exposure to Triclosan and Bisphenol Analogues B,F,P,S and Z in duplicated-diet Solid food of adults

In this paper, the authors present a study to estimate adults’ exposure to Triclosan and Bisphenol analogues, in their diet.

To perform this study, the authors supplied solid food to a cohort of 50 adults. The information regarding the type of solid food, the amount, the initial storage conditions and type of packaging is not provided.

This information should be added in the supplementary material. Moreover, it looks like the authors used the same preparation and extraction method (QUECHERS) for any type of food. Some study have shown that QUECHERS alone cannot remove some polar components that could interfere with the silylating reagent. Would the authors explain their choice of graphitized carbon black?

The solid food samples collected were stored in the freezer before homogenization. Would the authors explain whether they observed any influence of the thawing conditions?

Would the authors explain, the validation method, how they determine the limit of quantitation, the linearity of the method.  What spiking levels are used with the blanks, the matrix, and the samples?

The analysis were performed on a GCMS, the parameters regarding should be added in the supplementary material.

Data is acquired with a single quad mass spectrometer,  would the authors explain the data acquisition and the identification method used.

Given the ubiquitous presence of bisphenol A and bisphenol analogues in the laboratory, the authors should explain the precautions taken to control the background values.

The table, given in the supplementary material does not provide any valuable information and should be removed.

This paper is a follow-up of a previous study (ref 25) related to pyrethroid and lacks explanation regarding the design and the chemical analysis.

One reference should be added:

Contamination status of bisphenol A and its analogues (bisphenol S, F and B) in foodstuffs and the implications for dietary exposure on adult residents in Zhejiang Province

Jian Zhoua,b, Xiao-Hong Chena,b,⁎, Sheng-Dong Pana,b, Jun-Lin Wangc, Yi-Bin Zhengc,

Jiao-Jiao Xuc, Yong-Gang Zhaoa,b, Zeng-Xuan Caic, Mi-Cong Jina,b,

Author Response

Reviewer 5

Title: Exposure to Triclosan and Bisphenol Analogues B,F,P,S and Z in duplicated-diet Solid food of adults

In this paper, the authors present a study to estimate adults’ exposure to Triclosan and Bisphenol analogues, in their diet. To perform this study, the authors supplied solid food to a cohort of 50 adults.

Response: We did not supply/give any food items to the study participants. For clarification, we have added a new sentence to the second paragraph of section 2.2 of the Materials and Methods section: As part of their normal daily diets, the adult participants purchased their own food items during this study.

The information regarding the type of solid food, the amount, the initial storage conditions and type of packaging is not provided. This information should be added in the supplementary material.

Response: The food diary records showed that the 776 collected DDSF samples contained a total of 2,251 different solid food items consumed by the 50 participants over the six-week monitoring period. Also, in the food diaries the participants only estimated the amount of each solid food item (i.e., 2 cups of salad) they ate. Because of IRB constraints (human subject identifiers), we would only be able to provide a list of each solid food item and the estimated amount consumed (can’t be linked to a person). Therefore, we have elected not to include this very large file that would not add much value to the supplemental section of the manuscript. However, in section 2.2 of the Materials and Methods section, we have updated the following sentence with the recorded range of masses (g) for the food sampling bags: At this facility, a technician checked in all study items with each participant and recorded the food mass (g) of each sampling bag (range = 10.6 g – 1568 g) using a calibrated weight scale. For the initial storage conditions – In section 2.2. in the Materials and Methods section, we have revised the following sentence: The glass jars containing the homogenized food samples were stored in -20oC laboratory freezers until chemical analysis. Type of packaging – As the main purpose of the original Ex-R study was to assess adult exposures to pyrethroid insecticides in everyday environments, we did not collect any information on the types of food packaging.    

Moreover, it looks like the authors used the same preparation and extraction method (QUECHERS) for any type of food. Some study have shown that QUECHERS alone cannot remove some polar components that could interfere with the silylating reagent. Would the authors explain their choice of graphitized carbon black?

Response: Since these were composite duplicate diet solid food samples, the composition of each sample was unknown and varied by sample. This meant that for consistency, a single method was required for the extraction of all non-beverage food samples. No significant interference with the silylating reagent was observed. Only 5 of the 776 samples were affected. Graphitized carbon is a common sorbent used in dispersive solid phase extraction. In preliminary work using medium fat control diet, we evaluated recoveries and interferences using several pre-packaged sorbents, including C18, PSA, graphitized carbon, and mixes of these sorbents. Graphitized carbon alone demonstrated the best precision and accuracy among those tested.   

The solid food samples collected were stored in the freezer before homogenization. Would the authors explain whether they observed any influence of the thawing conditions?

Response:  We made an error when reporting that the DDSF food samples were placed into the laboratory freezers before being homogenized. The DDSF samples were homogenized before being placed into the laboratory freezers. We have revised two paragraphs in section 2.2 and 2.3 to provide the correct information as follows: Section 2.2 - At the laboratory, the DDSF samples (n=776) were homogenized individually using a vertical cutter mixer (R10-Ultra® or Robot Coupe R4N-D®) or a high-speed blender (WaringMBB518®) for high-liquid content foods like soups or fruit smoothies [25]. The samples were then aliquoted into 30 mL amber glass jars (12.0 g amounts each). The glass jars containing the homogenized food samples were stored in -20oC laboratory freezers until chemical analysis. Section 2.3. – At the laboratory, the 12.0 g aliquots for the DDSF samples (n=776) were removed from the freezers and thawed overnight in refrigerators. [25]……

Would the authors explain, the validation method, how they determine the limit of quantitation, the linearity of the method.  What spiking levels are used with the blanks, the matrix, and the samples?

Response: This method was evaluated for both accuracy and precision across the analytical concentration range of 0.5 to 500 ng/g in the medium fat control diet and at the 5 and 50 ng/g levels for both the low- and high-fat control diet. Based on the composition of all the field samples, it was assumed that the average fat content of the samples would generally fall near the medium fat level. For this reason, this method was evaluated with spiked samples prepared at 0.5 ng/g (n=6), 5 ng/g (n=3), 50 ng/g (n=3), and 500 ng/g (n=3) in medium fat control diet and at 5 ng/g (n=3) and 50 ng/g (n=3) in low and high fat control diets to determine the accuracy, precision and linearity across this concentration range. Also, three unfortified blank control diet samples were prepared for each type of control fat diet. In addition, sample aliquots were pulled from separate containers for each replicate. The residue levels evaluated were based on the estimated limit of quantitation (ELOQ) which was determined based on the lowest level that a standard solution could be accurately quantified on the GC/MS system. The ELOQ for most chemicals in this method was determined to be 0.5 ng/g. All samples, including blanks were fortified with internal standard solution at 30 ng/g as described in Section 2.3. We added in section 2.4 of the Materials and Methods that the matrix spikes and recovery spikes were fortified at 25 ng/g.

The analysis were performed on a GCMS, the parameters regarding should be added in the supplementary material.

Response: Parameters were added to the supplementary material (Table 1 and Table 2).

Data is acquired with a single quad mass spectrometer, would the authors explain the data acquisition and the identification method used.

Response: The mass selective detector (MSD) was operated in selected ion monitoring (SIM) mode with electron impact (EI) ionization. The ions monitored represent the silylated molecules where the hydroxyl groups are replaced by trimethylsilyl (TMS) groups. Excellent sensitivity and selectivity were achieved since the molecular ion plus TMS for each chemical were able to be used for detection and quantification. Detailed parameters for ions collected with retention times are described in the supplemental information.

 Given the ubiquitous presence of bisphenol A and bisphenol analogues in the laboratory, the authors should explain the precautions taken to control the background values.

Response: Supplies were evaluated for presence of BPA and analogues prior to use. In addition, reagent blanks were prepared with each sample batch in order to evaluate background levels that may be present in solvents and consumables as well as any potential contamination from equipment used during sample preparation. Glassware was also baked in a muffle furnace at 450º C for 8 hours prior to use (We have added this information to section 2.3 of the Materials and Methods section). 

The table, given in the supplementary material does not provide any valuable information and should be removed.

Response: As requested, Table 1 (Quality control samples) in Appendix A has been deleted from this manuscript and the text was updated to reflect this change in section 2.4 of the Materials and Methods section.

This paper is a follow-up of a previous study (ref 25) related to pyrethroid and lacks explanation regarding the design and the chemical analysis.

Response: No, the reviewer is incorrect.  We have provided the correct citation [24] (not 25) in the Introduction section: In the Ex-R study, BPA residue levels were found in 38% of the 776 duplicate-diet solid food (DDSF) samples (maximum = 138 ng/g) collected by 50 adults over a six-week monitoring period in North Carolina (NC) 2009–2011 [24] and in section 2.3 in the Materials and Methods section: “The six target phenols were concurrently extracted from each homogenized food sample using a QuEChERS (quick, easy, cheap, effective, rugged, and safe) method that was modified for complex food matrices [24,26,27].”  This citation [24] is Morgan et al. (2018) Distribution, variability, and predictors of urinary bisphenol A levels in 50 North Carolina Adults over a six-week monitoring period. Env. Int. 2018, 112, 85–99. This published articled provided an in-depth discussion of the sampling methodology and chemical analysis for bisphenol A in the 776 DDSF samples from the Ex-R study – which are the same food samples and methodology that was used for the target bisphenols and triclosan in this current paper.

One reference should be added:

Contamination status of bisphenol A and its analogues (bisphenol S, F and B) in foodstuffs and the implications for dietary exposure on adult residents in Zhejiang Province

Jian Zhoua,b, Xiao-Hong Chena,b,, Sheng-Dong Pana,b, Jun-Lin Wangc, Yi-Bin Zhengc,

Jiao-Jiao Xuc, Yong-Gang Zhaoa,b, Zeng-Xuan Caic, Mi-Cong Jina,b,

Response: As recommended by the reviewer, we have added this important citation to the manuscript.

Round 2

Reviewer 2 Report

The Authors replied adequately to my questions and concerns, and made appropriate changes. Therefore the paper can be recommended for publication

Reviewer 3 Report

I respect your responses and accept this version of the manuscript.

Reviewer 4 Report

The authors answered the questions and added some clarifications in the paper.

Line 161, the exact composition of the internal standard solution should be added

Besides some minor typo errors, I believe this paper can be published in its present form